# ExpNote: Black-box Large Language Models are better Task Solvers with Experience Notebook

**Wangtao Sun**[1,2], **Xuanqing Yu**[2,3], **Shizhu He**[1,2], **Jun Zhao**[1,2], **Kang Liu**[1,2,4]

[1]*The Laboratory of Cognition and Decision Intelligence for Complex Systems,*
*Institute of Automation, Chinese Academy of Sciences, Beijing, China*
[2]*School of Artificial Intelligence, University of Chinese Academy of Sciences, Beijing, China*
[3]*CAS Engineering Laboratory for Intelligent Industrial Vision,*
*Institute of Automation, Chinese Academy of Sciences, Beijing, China*
[4]*Shanghai Artificial Intelligence Laboratory*
{sunwangtao2021, yuxuanqing2021}@ia.ac.cn
{shizhu.he, jzhao, kliu}@nlpr.ia.ac.cn

## Abstract

Black-box Large Language Models (LLMs) have shown great power in solving various tasks and are considered general problem solvers. However, LLMs still fail in many specific tasks although understand the task instruction. In this paper, we focus on the problem of boosting the ability of black-box LLMs to solve downstream tasks. We propose Exp-Note, an automated framework to help LLMs better adapt to unfamiliar tasks through reflecting and noting experiences from training data and retrieving them from external memory during testing. We evaluate ExpNote on multiple tasks and the experimental results demonstrate that the proposed method significantly improves the performance of black-box LLMs. The data and code are available at https://github.com/forangel2014/ExpNote.

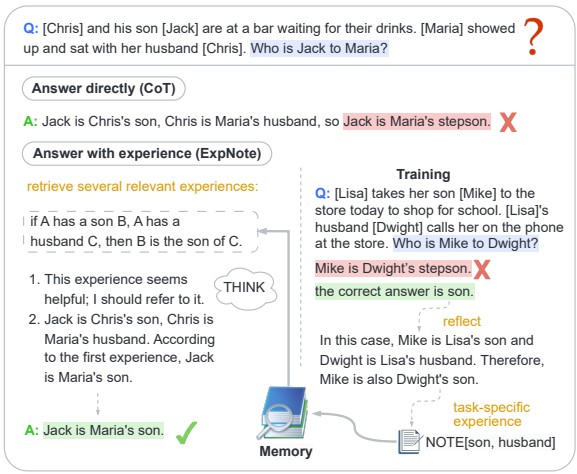

Figure 1: An illustration of how ExpNote assists LLM in enhancing the effectiveness of task-solving. ExpNote can automatically generalize relevant experiences from other samples and apply them to specific tasks.

## 1 Introduction

Large Language Models (LLMs) have demonstrated astonishing capabilities in natural language understanding and generation (Wei et al., 2022; Huang et al., 2022; Sun et al., 2022; Bang et al., 2023). However, due to the limited parameters and context processing length, LLMs are not able to master all task-specific knowledge in real-world applications. As a result, LLMs may perform mediocre on some specific tasks, such as inductive reasoning (Bang et al., 2023) and entity recognition (Chen et al., 2023).

Therefore, how to make LLMs adapt to the downstream tasks has tracked more and more attention. Recent techniques such as prefix-tuning (Li and Liang, 2021), P-tuning (Liu et al., 2021) and LoRA (Hu et al., 2021) proposed low-cost solutions for fine-tuning LLMs. However, these methods are not capable of black-box powerful LLMs, such as ChatGPT and GPT4 (OpenAI, 2023).

To empower black-box LLMs on specific tasks, several works (Madaan et al., 2022; Dalvi et al., 2022) have focused on equipping the LLMs with external dynamic memory to store useful task-specific knowledge and facts. However, these task-specific knowledge and facts in the memory usually come from expert annotation or human feedback, which is very costly to obtain. On the other hand, several researchers (Shinn et al., 2023; Madaan et al., 2023; Akyürek et al., 2023) try to exploit the reflection ability of LLMs to automatically generate such knowledge for specific tasks. However, most reflection-based methods are only able to empower the LLMs in the same case, without the ability to generalize to other instances.

Thus, this paper proposes a framework Exp-Note (Experience Notebook), to empower black-box LLMs on downstream tasks by learning and using task-specific experience automatically. We

equip the LLMs with a dynamic memory and design several commands to help LLMs interact with it. In specific, in the training stage, ExpNote guides LLMs to generate task-specific experiences and store them in an external memory. In the testing stage, ExpNote uses a retriever to retrieve relevant experiences from the memory, the learned experiences will help the LLMs to solve the cases which they failed to answer directly (Figure 1).

We evaluate ExpNote on multiple tasks. The results show that ExpNote can empower the LLMs effectively, and significantly outperform other prompting methods like ordinary in-context learning (CoT, Wei et al. 2022), memory-based method (TeachMe, Dalvi et al. 2022), and case-by-case reflection-based method (Reflexion, Shinn et al. 2023).

Moreover, we empirically compared different types of experiences to examine their effectiveness in helping LLMs adapt to unfamiliar tasks. Specifically, we compared learned task-specific experiences with original cases and experiences learned from positive cases (succeeded) and negative cases (failed). We find that prompting with experiences is more helpful than original cases for LLMs to generalize to new cases, and experiences from both positive and negative cases are beneficial.

The major contributions of this paper are two-fold:

- We propose a framework ExpNote to empower the LLMs in various tasks through interacting with dynamic memory. ExpNote conducts fully automated reflection, noting, and retrieval, without the need of any annotated knowledge and facts or any human feedback.
- We investigate different types of experiences and show the learned task-specific experiences help LLMs to better generalize than the original cases in the task, and experiences from both positive and negative cases are beneficial.

## 2   Related Work

### 2.1   Language Model Taking Action

In order to address the lack of knowledge, reasoning, and specific abilities in large models, many efforts have focused on utilizing external knowledge sources and tools to assist large models in completing tasks. Toolformer (Schick et al., 2023) proposed fine-tuning on API-augmented datasets to enable the LLMs to master the ability to use external tools with API calls, thereby improving the performance of the LLMs in a series of tasks. ReAct (Yao et al., 2022) proposed that by using a Wikipedia search API and generating trajectories similar to human thinking, the LLMs can utilize external knowledge during reasoning and provide interpretable reasoning paths. HuggingGPT (Shen et al., 2023) proposes to solve any AI task by using the models on the huggingface as its toolkit.

### 2.2   Language Model with Dynamic Memory

Some existing works have noticed the need to equip LLMs with dynamic memory. MemPrompt (Madaan et al., 2022) retrieves the stored user feedback of the intention for similar questions to enhance the current prompt for the LLMs. TeachMe (Dalvi et al., 2022) allows LLMs to store the missing and wrong facts during the QA task with the correction of user feedback. These methods created a new paradigm to boost the ability of LLMs in a general way. However, they rely heavily on human feedback or annotated facts. REMEMBERER (Zhang et al., 2023) proposed to consider LLMs as a semi-parametric RL agent. It trains LLMs to take the next action based on the retrieved (observation, action, Q-value) tuple.

### 2.3   Language Model Reflection

Recently, some works have been proposed to correct the mistakes of LLMs in conducting specific tasks by using their capacity of self-reflection. Reflexion (Shinn et al., 2023) focused on sequential decision-making tasks. A heuristic function is adopted to judge whether the trial is successful or not. And LLMs will reflect those trials that are thought to have failed. The reflective information will be used to support the LLMs in improving their own decision-making process in the next trial. Self-refine (Madaan et al., 2023) proposed a method to iteratively improve the output of a large model through its own feedback, achieving improvements in multiple generation tasks. However, these reflection methods are limited to certain cases, without being abstract and able to generalize to other data points.

### 2.4   Language Model Thought Chain

Furthermore, there have been some efforts to improve the reasoning performance of LLMs by enhancing their thought chains in specific tasks. For example, DIVERSE (Li et al., 2023) proposed a method that generates multiple different reasoning paths and uses a validator for weighted voting to

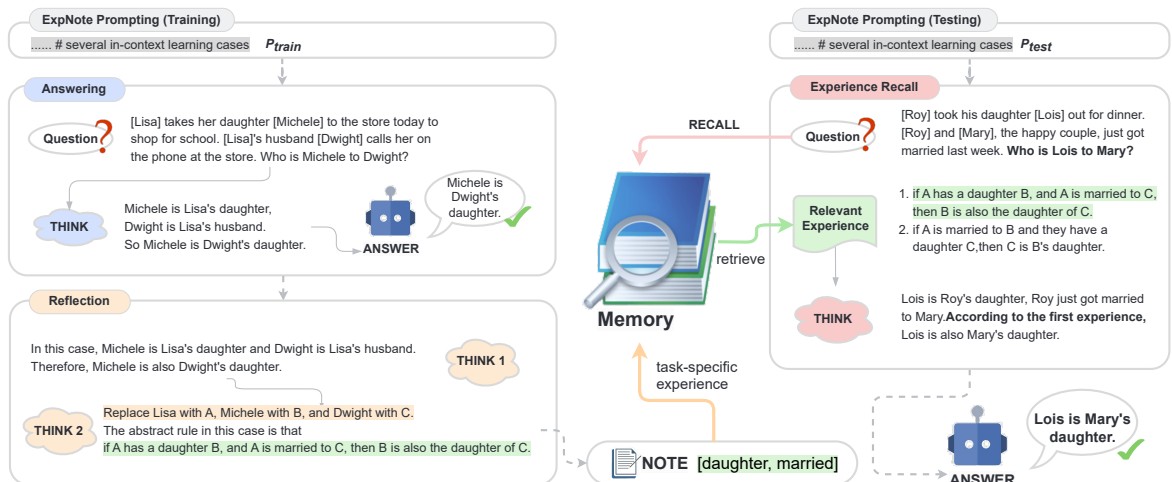

Figure 2: The framework of ExpNote. This framework shows how LLMs use ExpNote to solve specific tasks, including the training (left) and testing (right) stages.

filter out incorrect answers. However, this method demands manual construction of reasoning paths for each training question-answer pair and extensive human evaluation, restricting its use on large datasets.

Drozdov (Drozdov et al., 2022) and others introduced a technique that decomposes complex questions into sub-questions and provides answers to these sub-questions, serving as hints to assist the model in reaching the final answer. Faithful CoT (Lyu et al., 2023) , on the other hand, prompts a language model to translate complex queries into a reasoning chain that includes question decomposition and corresponding symbolic language solving, thus enhancing interpretability.

These approaches offer intriguing ideas for improving the reasoning performance of LLMs but still face challenges related to the need for substantial high-quality annotations, difficulties in reusing experiences, and sample generalization.

## 3 ExpNote

### 3.1 The Framework

As shown in Figure 2, all the tasks are formalized as the tuple $(x, y)$, where $x$ is the input question and $y$ is the desired answer. For each task, we write prompts $P_{train}$ and $P_{test}$ that encourage the LLM to use ExpNote following the illustrations. In the training stage, LLM is first ordered to infer the answer like ordinary CoT reasoning.

$$\hat{y} \sim p_{LLM}(\cdot|P_{train}, x) \qquad (1)$$

After the answer is obtained, the ExpNote will produce feedback $F(\hat{y}, y)$ to LLM depending on whether $\hat{y} = y$. Note that this feedback $F(\hat{y}, y)$ only includes a simple prompt containing the ground-truth of the current question, without any additional knowledge like TeachMe (Dalvi et al., 2022). Then LLM is supposed to reflect and store the learned experience $e$ into the memory.

$$e \sim p_{LLM}(\cdot|P_{train}, x, F(\hat{y}, y)) \qquad (2)$$

Where $e$ is a key-value pair of learned task-specific experience, e.g. key = *daughter, married* and value = *if A has a daughter B, and A is married to C, then B is also the daughter of C* (Figure 2). This process is achieved by taking $n$ extra actions to interact with the memory, which we will describe in Sec 3.2.

In the testing stage, ExpNote will use the testing instance as the search query to retrieve $k$ experiences from the dynamic memory. The retrieved experiences will be added to the prompts for the LLM. Then LLM will decide whether to refer to these experiences and finally output the answer.

$$\{e_i\}_{i=1}^k = Retrieve(x, k)$$
$$\hat{y} \sim p_{LLM}(\cdot|P_{test}, x, \{e_i\}_{i=1}^k) \qquad (3)$$

The full examples of ExpNote on different tasks are shown in Appendix D.

### 3.2 Interaction Commands

ExpNote designs several commands for LLMs to interact with the memory, including summarizing and applying the experiences (**THINK**), storing the

experiences in the memory (**NOTE**), and retrieving relevant experiences for the testing instances (**RECALL**). They are described in detail as follows:

- **THINK[arg]**. Inspired by ReAct (Yao et al., 2022), in both training and testing stages, we enable LLM to use command **THINK** to organize its current thoughts and make the next decision.
- **NOTE[arg1]: arg2**. In the training stage, we prompt the LLMs to use this command **NOTE** after answering each question. The command **NOTE** will store the experience **arg2** as a value in the memory with its key **arg1**.
- **RECALL[arg]**. In the testing stage, this command is automatically executed by ExpNote at the beginning of each question to recall relevant experiences. ExpNote will use a retriever to retrieve up to $k$ relevant experiences using the question **arg** as the search query. The content of these experiences will then be added to the prompt.

## 4 Experiments

In this section, we want to answer the following research questions:

- **RQ1**. Is ExpNote able to help LLMs to adapt to new tasks effectively?
- **RQ2**. Which kinds of experiences help LLMs solve tasks better?

### 4.1 Datasets

To show the effectiveness of ExpNote in handling various tasks, we select multiple datasets from different tasks for empirical study, including CLUTRR (inductive reasoning, Sinha et al. 2019), METS-CoV (medical entity recognition, Zhou et al. 2022), EMOJI (text-to-emoji prediction, Felbo et al. 2017). Besides, we propose a dataset LETS (letter splicing) to evaluate the symbolic reasoning capability of LLM enhanced with ExpNote. The detail of LETS can be found in Appendix A. For each task, we tested ExpNote and other methods on 100 cases. The other setups of the experiments can be found in Appendix B.

### 4.2 Baselines

To answer the **RQ1**, apart from basic zero-shot and few-shot settings, we select the ordinary in-context learning method (CoT, Wei et al. 2022), memory-based method (TeachMe, Dalvi et al. 2022) and case-by-case reflection-based method (Reflexion, Shinn et al. 2023) for comparison.

| methods | CLUTRR | METS | EMOJI | LETS |
|---------|--------|------|-------|------|
| zero-shot | 36 | 23 | 34 | 35 |
| few-shot | 44 | 61 | 47 | 1 |
| CoT | 40 | 54 | 54 | 60 |
| TeachMe | 31 | 51 | 56 | 56 |
| Reflexion | 54 | 62 | 71 | 68 |
| ExpNote | **61** | **66** | **74** | **89** |

Table 1: Accuracy of ExpNote and baselines on 4 datasets.

| variants | CLUTRR | METS | EMOJI | LETS |
|----------|--------|------|-------|------|
| *disabled* | 35 (0) | 49 (0) | 57 (0) | 50 (0) |
| *case* | 49 (128) | 59 (279) | 58 (20) | 51 (87) |
| *positive* | 51 (73) | 56 (166) | 66 (14) | 64 (41) |
| *negative* | 55 (55) | 52 (113) | 60 (6) | 71 (46) |
| ExpNote | **61** (128) | **66** (279) | **74** (20) | **89** (87) |

Table 2: Accuracy of ExpNote and its variants on 4 datasets. The numbers in the small bracket are numbers of experiences stored in the memory.

- CoT (Wei et al., 2022): Several cases of solving the task using Chain-of-Thought are shown to the LLM.
- TeachMe (Dalvi et al., 2022): As the core facts or human feedback of these specific tasks are hard to obtain, we adopt a commonsense knowledge base, Conceptnet (Speer et al., 2017), to serve as the memory for TeachMe.
- Reflexion (Shinn et al., 2023): As the heuristic function of repetitive action detection described in Reflexion is not working for these tasks. Thus we allow Reflexion to do a little cheating: it is allowed to try again after every failed trial without being informed of the ground-truths. This setting is equivalent to obtaining a golden function that accurately determines the success/failure of each trial with a 100% success rate.

To answer the **RQ2**, we have also implemented several variants of ExpNote:

- *disabled*: This variant adopts the reasoning form of ExpNote while disabling its retrieval function.
- *case*: This variant will recall the original questions and answers of the noted cases instead of the learned experiences.
- *positive* / *negative*: This variant only retains the experiences that learned from the training sample which LLMs answered correctly / incorrectly.

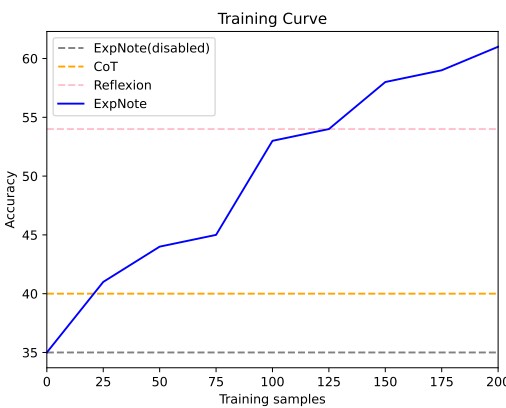

Figure 3: The training curve in CLUTRR dataset.

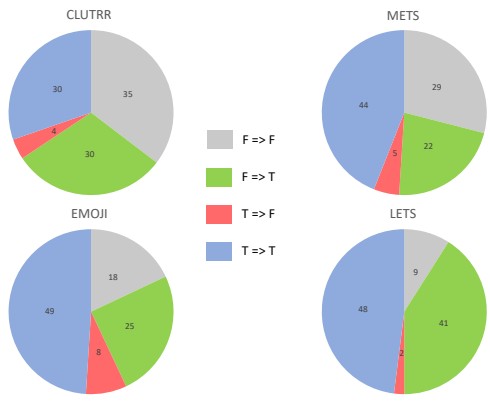

Figure 4: Improvement analysis of ExpNote on 4 datasets.

## 4.3 Results

As shown in Table 1, the full ExpNote method achieved the best performance on all datasets, 20.5% higher on average than the CoT method. TeachMe failed to outperform few-shot, as task-specific knowledge is hard to obtain without human feedback. Compared with Reflexion, note that even if we make Reflexion cheat to identify the failed trial with 100% success rate, it still falls behind ExpNote.

Compared with other variants of ExpNote, *disabled* retrieves no experience in the testing stage, thus degrading the performance of CoT (even worse) as expected. We also discovered that *case* performs worse than full ExpNote although retrieving exactly the same cases for all of 4 tasks. We can then conclude that abstract knowledge or rules are more capable of helping LLMs to generalize to testing cases. Moreover, *positive* and *negative* both fall behind the full ExpNote while still outperforming baselines. We made an efficiency analysis in Appendix C and the results show that experiences from both positive and negative cases are more efficient than the other respectively on two datasets. These results indicated that experiences learned from both positive cases and negative cases are useful for LLM to generalize to test sets.

We also observed the performance changes of the model with the number of training samples. As shown in Figure 3, in CLUTRR, ExpNote starts from training with 0 samples (equivalent to *disabled*) and ends with training with 200 samples. The performance of ExpNote on the testing set continually grows with the number of training samples, showing that ExpNote continually learns new knowledge during the training stage.

## 4.4 Improvement Analysis

We also analyze how many cases are corrected by introducing experiences in each dataset. As shown in Figure 4, we plot the distribution of cases in 4 conditions:

- F => F: a case is originally answered incorrectly in *disabled* and also answered incorrectly with ExpNote.
- F => T: a case is originally answered incorrectly in *disabled* but answered correctly with ExpNote.
- T => T: a case is originally answered correctly in *disabled* and also answered correctly with ExpNote.
- T => F: a case is originally answered correctly in *disabled* but answered incorrectly with ExpNote.

In Figure 4, we demonstrate that ExpNote helps LLMs correct a certain amount of errors (the green part) at the cost of producing a few new errors (the red part) in all 4 datasets. And we can observe around 50% incorrect answers in *disabled* (gray + green) are corrected (green) with ExpNote.

## 5 Conclusion

In this paper, we propose ExpNote, an automated framework to help black-box LLMs adapt to specific downstream tasks by interacting with dynamic memory. We carried out experiments on multiple datasets from different tasks and showed that ExpNote can effectively improve the performance of LLMs better than other prompting methods. We also found that the learned task-specific experiences help LLMs to better generalize than the original cases in the task, and experiences learned from both positive cases and negative cases are valuable.

## Limitations

Although ExpNote is able to empower the LLMs in various tasks, it may be less effective on these case-by-case tasks, like summarizing or creative writing. In these tasks, the cases share little common knowledge or rules, which makes it hard for ExpNote to help LLMs generalize.

## Ethics Statement

This paper proposes a method for augmenting black-box LLMs. All experiments are conducted on publicly available datasets. Thus there is no data privacy concern. Meanwhile, this paper does not involve human annotations, and there are no related ethical concerns.

## Acknowledgements

This work was supported by the National Key R&D Program of China (2022ZD0160503) and the National Natural Science Foundation of China (No.62376270No.61831022). This work was supported by the Strategic Priority Research Program of Chinese Academy of Sciences (No.XDA27020100), Youth Innovation Promotion Association CAS and OPPO Research Fund.

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

## A  LETS Dataset

As existing symbolic reasoning datasets, such as word sorting in BIG-bench (Srivastava et al., 2022), are designed to test the zero-shot reasoning ability of LLMs and always lack a training set, we therefore propose the LETS, a similar symbolic reasoning dataset while enabling LLMs to learn and generalize.

LETS require the language model to splice the letters at a given index of several words together. For example, given the query *Splice the 5th letter of "sleep", the 2nd letter of "official", and the 5th letter of "neglect" together*, the model is supposed to output *pfe* as the answer.

We randomly select 100 words with lengths of 4-10 as the vocabulary. To generate the training and testing set, for each instance, we randomly picked 3 different words from the vocabulary and randomly selected their indexes.

## B  Setup

For the LLM, we use ChatGPT (gpt-3.5-turbo) via Openai API calls.

For each task, due to the size limitations of the datasets themselves, we test all methods on 100 testing cases. In fact, a large amount of related work is also tested using samples of similar magnitude, such as TeachMe (OBQA, 500, Dalvi et al. 2022), ReAct (ALFWorld, 134; WebShop, 500, Yao et al. 2022), Reflexion (consistent with ReAct, Shinn et al. 2023). Considering Expnote will interact with the environment multiple turns for a single case, the actual number of generations for LLMs can be 4 to 5 times higher. And We adopt a minimal training set with it size 2:1 to the testing set (and 1:1 in EMOJI and LETS datasets).

For all ExpNote variants, we write 2-3 ExpNote usage cases for the LLM as few-shot prompting; we choose $n = 4$ for training (the LLM is able to take 4 extra actions to **THINK** and **NOTE** after obtaining the ground-truth of each case), and $n = 0$ for testing (the LLM is not able to access to the ground-truth).

For the retriever, we implemented a word-based retriever to retrieve experience by matching words in the query and the key of experience, and it retrieves up to $k = 3$ experiences for each case in the testing stage. When ExpNote fails to retrieve relevant experience, a failure prompt "No relevant experience" will be returned to the LLM.

## C  Effciency Analysis

We can define the efficiency of each type of experience as

$$\text{eff}(type) = \frac{\text{Perf}(type) - \text{Perf}(disabled)}{\text{Cnt}(type)} \quad (4)$$

where $type$ refers to positive or negative, Perf() represents the corresponding variant's performance in Table 2, and Cnt($type$) represents the number of experiences of that type. Then we can calculate the efficiency of positive and negative experiences based on the data in Table 2.

As shown in Table 3, experiences from both positive and negative cases are more efficient than the

| type | CLUTRR | METS | EMOJI | LETS |
|---|---|---|---|---|
| *positive* | 0.219 | **0.042** | **0.643** | 0.341 |
| *negative* | **0.364** | 0.026 | 0.500 | **0.456** |

Table 3: Efficiency of Positive and Negative Experiences on 4 datasets.

other on two datasets. These results indicated that experiences learned from both positive cases and negative cases are useful for LLM to generalize to test sets.

## D    Full Example

## Example Trajectory - Training and Testing Cases on the CLUTRR Dataset

| | Training Stage | Testing Stage |
|---|---|---|
| **ExpNote Demonstrations** | SYSTEM: You are a smart assistant and You need to complete the specified task by talking to the SYSTEM. The dialog between you and SYSTEM will be shown in the form "ASSISTANT:" and "SYSTEM:".

SYSTEM: Your task is to answer the question given a short story.

SYSTEM: Here are several examples.
SYSTEM: case 1.
SYSTEM: [Amy] is having a meal with her father [Jackson]. [Lucy] is happy watching TV with her husband [Jackson]. Who is Amy to Lucy?
ASSISTANT: **THINK**[Amy is Jackson's daughter, Jackson is Lucy's husband. Amy is Lucy's husband's daughter. so Amy is Lucy's daughter-in-law.]
SYSTEM: OK.
ASSISTANT: **ANSWER**[Amy is Lucy's daughter-in-law.]
SYSTEM: Your answer is daughter-in-law, however, the correct answer is daughter.
ASSISTANT: **THINK**[In this case, Amy has a father Jackson, Jackson is the husband of Lucy. The conclusion is that Amy is the daughter of Lucy.]
SYSTEM: OK.
ASSISTANT: **THINK**[Replace Amy with A, Jackson with B, and Lucy with C. The abstract rule in this case is that if A has a father B, B is the husband of C, then A is the daughter of C.]
SYSTEM: OK.
ASSISTANT: **NOTE**[father, husband]: if A has a father B, B is the husband of C, then A is the daughter of C.
SYSTEM: case 2.
SYSTEM: [Chris] and his son [Jack] are at a bar waiting for their drinks. [Maria] showed up and sat with her husband [Chris]. Who is Jack to Chris?
ASSISTANT: **THINK**[Jack is Chris's son, Chris is Maria's husband. Jack is Maria's husband's son. so Jack is Maria's stepson.]
SYSTEM: OK.
ASSISTANT: **ANSWER**[Jack is Maria's stepson.]
SYSTEM: Your answer is stepson, however, the correct answer is son.
ASSISTANT: **THINK**[In this case, Jack is the son of Chris, Chris is the husband of Maria. The conclusion is that Jack is the son of Maria.]
SYSTEM: OK.
ASSISTANT: **THINK**[Replace Jack with A, Chris with B, and Maria with C. The abstract rule in this case is that if A is the son of B, B is the husband of C, then A is the son of C.]
SYSTEM: OK.
ASSISTANT: **NOTE**[son, husband]: if A is the son of B, B is the husband of C, then A is the son of C.
SYSTEM: case 3.
SYSTEM: [George] drove her daughter [Amy] to the hospital. [Amy]'s son [Bob] showed up shortly after. Who is Bob to George?
ASSISTANT: **THINK**[Amy is George's daughter, Bob is Amy's son. Bob is George's daughter's son. so Bob is George's grandson.]
SYSTEM: OK.
ASSISTANT: **ANSWER**[Bob is George's grandson.]
SYSTEM: Your answer is correct.
ASSISTANT: **THINK**[In this case, George has a daughter Amy, Amy has a son Bob. The conclusion is that Bob is the grandson of George.]
SYSTEM: OK.
ASSISTANT: **THINK**[Replace George with A, Amy with B, and Bob with C. The abstract rule in this case is that if A has a daughter B, B has a son C, then C is the grandson of A.]
SYSTEM: OK.
ASSISTANT: **NOTE**[daughter, son]: if A has a daughter B, B has a son C, then C is the grandson of A. | SYSTEM: You are a smart assistant and You need to complete the specified task by talking to the SYSTEM. The dialog between you and SYSTEM will be shown in the form "ASSISTANT:" and "SYSTEM:".

SYSTEM: Your task is to answer the question given a short story.

SYSTEM: Here are several examples.
SYSTEM: case 1.
SYSTEM: [Amy]'s brother [Mike] and her went to get ice cream. [Mike] is the proud father of the lovely [Jackson]. Who is Amy to Jackson?
SYSTEM: No relevant experience.
ASSISTANT: **THINK**[Seems there is no relevant experience about this question, I need to rely on myself]
SYSTEM: OK.
ASSISTANT: **THINK**[Mike is Amy's brother, Amy is female, Mike is Jackson's father. Dorophy is Jackson's father's sister. so Amy is Jackson's aunt.]
SYSTEM: OK.
ASSISTANT: **ANSWER**[Amy is Jackson's aunt.]
SYSTEM: Your answer is correct.
SYSTEM: case 2.
SYSTEM: [Chris] and his son [Jack] are at a bar waiting for their drinks. [Maria] showed up and sat with her husband [Chris]. Who is Jack to Maria?
SYSTEM: Here are several possible relevant experience:
SYSTEM: if A has a son B, A has a wife C, then B is the son of C.
ASSISTANT: **THINK**[This experience seems helpful, I should refer to it.]
SYSTEM: OK.
ASSISTANT: **THINK**[Jack is Chris's son, Chris is Maria's husband. According to this experience, Jack is Maria's son.]
SYSTEM: OK.
ASSISTANT: **ANSWER**[Jack is Maria's son.]
SYSTEM: Your answer is correct.
SYSTEM: case 3.
SYSTEM: [George] drove her daughter [Amy] to the hospital. [Amy]'s son [Bob] showed up shortly after. Who is Bob to George?
SYSTEM: Here are several possible relevant experience:
SYSTEM: if A has a daughter B, A has a son C, then B is the brother of C.
ASSISTANT: **THINK**[This experience doesn't seem to be helpful in answering this question. I need to rely on myself.]
SYSTEM: OK.
ASSISTANT: **THINK**[Amy is George's daughter, Bob is Amy's son. Bob is George's daughter's son. so Bob is George's grandson.]
SYSTEM: OK.
ASSISTANT: **ANSWER**[Bob is George's grandson.]
SYSTEM: Your answer is correct. |
| **Interactive Trajectories** | SYSTEM: Now here is the question:
SYSTEM: [Dan]'s brother, [Micheal], and his grandmother, [Gabrielle], met him for his birthday dinner at the local diner. Who is Micheal to Gabrielle?
ASSISTANT: **THINK**[Micheal is Dan's brother, Gabrielle is Dan's grandmother. Micheal is Dan's grandmother's grandson. so Micheal is Gabrielle's grandson.]
SYSTEM: OK.
ASSISTANT: **ANSWER**[Micheal is Gabrielle's grandson.]
SYSTEM: Your answer is correct.
ASSISTANT: **THINK**[In this case, Dan's brother is Micheal and Gabrielle is Dan's grandmother. Therefore, Micheal is Gabrielle's grandson.]
SYSTEM: OK.
ASSISTANT: **THINK**[Replace Dan with A, Micheal with B, and Gabrielle with C. The abstract rule in this case is that if A's brother is B and C is A's grandmother, then B is C's grandson.]
SYSTEM: OK.
ASSISTANT: **NOTE**[brother, grandmother]: if A's brother is B and C is A's grandmother, then B is C's grandson.
SYSTEM: OK.
ASSISTANT: END | SYSTEM: Now here is the question:
SYSTEM: [Vincent]'s brother, [Dan], and his grandmother, [Gabrielle], met him for his birthday dinner at the local diner. Who is Dan to Gabrielle?
SYSTEM: Here are several possible relevant experience:
SYSTEM: if A's brother is B and C is A's grandmother, then B is C's grandson.
ASSISTANT: **THINK**[This experience seems helpful, I should refer to it.]
SYSTEM: OK.
ASSISTANT: **THINK**[Dan is Vincent's brother, Gabrielle is Vincent's grandmother. According to this experience, Dan is Gabrielle's grandson.]
SYSTEM: OK.
ASSISTANT: **ANSWER**[Dan is Gabrielle's grandson.]
SYSTEM: Your answer is correct. |

Figure 5: The example trajectories of ExpNote on the CLUTRR Dataset. The left part is a training case while the right part is a corresponding testing case using that training experience. The blue parts are the ExpNote demonstrations (prompts $P_{train}/P_{test}$). The yellow parts are the interactive trajectories between LLM and ExpNote. The sentence highlighted in green is the learned experience.

## Example Trajectory - Training and Testing Cases on the METS Dataset

| | Training Stage | Testing Stage |
|---|---|---|
| **ExpNote Demonstrations** | SYSTEM: You are a smart assistant and You need to complete the specified task by talking to the SYSTEM. The dialog between you and SYSTEM will be shown in the form "ASSISTANT:" and "SYSTEM:".

SYSTEM: Your task is to identify the exactly one entity of type vaccine, disease, drug, or symptom in the given text. Returns in the format of (entity, type).

SYSTEM: case 1.
SYSTEM: Identify an entity of type vaccine, disease, drug, or symptom in this text. Returns in the format of (entity, type). Do not include "covid" in the answer.
@celliottability @fordnation @jkwan_md @BrooksFallis I am devastated beyond words . Just found out that my son tested positive for covid . He had a transplant last May . So far he is just nauseous but we all know that with the variants that could change on a dime .
ASSISTANT: **ANSWER**[(transplant, symptom)]
SYSTEM: Your answer is (transplant, symptom), however, the correct answer is (nauseous, symptom).
ASSISTANT: **THINK**[I made a mistake, transplant is not the desired entity, and nauseous is a symptom.]
SYSTEM: OK.
ASSISTANT: **NOTE**[transplant]: "transplant" is not a desired entity.
SYSTEM: OK.
ASSISTANT: **NOTE**[nauseous]: "nauseous" is a symptom.
SYSTEM: OK.
SYSTEM: case 2.
SYSTEM: Identify an entity of type vaccine, disease, drug, or symptom in this text. Returns in the format of (entity, type). Do not include "covid" in the answer.
@yeahrightgirlhg Except it absolutely is . mRNA vaccines have been around since the early 90 ' s and are well researched . Meanwhile you know the real risks of Covid and to argue that the vaccine is more dangerous than the virus just has no basis in reality .
ASSISTANT: **ANSWER**[(vaccine, vaccine)]
SYSTEM: Your answer is (vaccine, vaccine), however, the correct answer is (mRNA vaccines, vaccine).
ASSISTANT: **THINK**[I made a mistake, vaccine is not the desired entity, and mRNA vaccines is a vaccine.]
SYSTEM: OK.
ASSISTANT: **NOTE**[vaccine]: "vaccine" is not a desired entity.
SYSTEM: OK.
ASSISTANT: **NOTE**[mRNA vaccines]: "mRNA vaccines" is a vaccine.
SYSTEM: OK. | SYSTEM: You are a smart assistant and You need to complete the specified task by talking to the SYSTEM. The dialog between you and SYSTEM will be shown in the form "ASSISTANT:" and "SYSTEM:".

SYSTEM: Your task is to identify the exactly one entity of type vaccine, disease, drug, or symptom in the given text. Returns in the format of (entity, type).

SYSTEM: Here are several examples.
SYSTEM: case 1.
SYSTEM: Identify an entity of type vaccine, disease, drug, or symptom in this text. Returns in the format of (entity, type). Do not include "covid" in the answer.
hot take : coronavirus is the sniffles.
SYSTEM: No relevant experience.
ASSISTANT: **THINK**[Seems there is no relevant experience about this question, I need to rely on myself.]
SYSTEM: OK.
ASSISTANT: **THINK**[sniffles seems to be a symptom, so the answer is (sniffles, symptom)]
SYSTEM: OK.
ASSISTANT: **ANSWER**[(sniffles, symptom)]
SYSTEM: case 2.
SYSTEM: Identify an entity of type vaccine, disease, drug, or symptom in this text. Returns in the format of (entity, type). Do not include "covid" in the answer.
@pattpattpattyo @AaronSuttles @fluffopotamus88 Pneumonia due to COVID just got intubated check up on @RandRCigars for updates as well.
SYSTEM: Here are several possible relevant experience:
SYSTEM: "pneumonia" is a disease.
SYSTEM: "COVID" is not a desired entity.
ASSISTANT: **THINK**[Based on these experiences, "pneumonia" is a disease, "COVID" is not a desired entity. So the answer is (pneumonia, disease).]
SYSTEM: OK.
ASSISTANT: **ANSWER**[(pneumonia, disease)] |
| **Interactive Trajectories** | SYSTEM: Now here is the question:
SYSTEM: Identify an entity of type vaccine, disease, drug, or symptom in this text. Returns in the format of (entity, type). Do not include "covid" in the answer.
@nealejones So many sick people passing as normal . Like a Covid specific PTSD from the initial shock of it.
ASSISTANT: ANSWER[(PTSD, symptom)]
SYSTEM: Your answer is (PTSD, symptom), however, the correct answer is (PTSD, disease).
ASSISTANT: **THINK**[I made a mistake, PTSD is a disease, not a symptom.]
SYSTEM: OK.
ASSISTANT: **NOTE**[PTSD]: "PTSD" is a disease.
SYSTEM: OK.
ASSISTANT: **NOTE**[symptom]: "symptom" is not a desired entity.
SYSTEM: OK.
ASSISTANT: END | SYSTEM: Now here is the question:
SYSTEM: Identify an entity of type vaccine, disease, drug, or symptom in this text. Returns in the format of (entity, type). Do not include "covid" in the answer.
@ThatAmish1 @PolitiBunny How soon before that dumbass Fauci claims flatulence is a symptom of COVID and we ' ll need to insert butt plugs to prevent expelling the malodorous airbone " infectious " particles ?.
SYSTEM: Here are several possible relevant experience:
SYSTEM: "symptom" is not a desired entity.
ASSISTANT: **THINK**[Based on these experiences, "symptom" is not a desired entity. So the answer is (flatulence, symptom).]
SYSTEM: OK.
ASSISTANT: **ANSWER**[(flatulence, symptom)]
SYSTEM: Your answer is correct. |

Figure 6: The example trajectories of ExpNote on the METS Dataset. The left part is a training case while the right part is a corresponding testing case using that training experience. The blue parts are the ExpNote demonstrations (prompts $P_{train}/P_{test}$). The yellow parts are the interactive trajectories between LLM and ExpNote. The sentence highlighted in green is the learned experience.

## Example Trajectory - Training and Testing Cases on the EMOJI Dataset

| | Training Stage | Testing Stage |
|---|---|---|
| **ExpNote Demonstrations** | SYSTEM: You are a smart assistant and You need to complete the specified task by talking to the SYSTEM. The dialog between you and SYSTEM will be shown in the form "ASSISTANT:" and "SYSTEM:".

SYSTEM: Your task is to select the emoji that best matches the text.

SYSTEM: Here are several examples.
SYSTEM: case 1.
SYSTEM: When someone stole my bike. \U0001f616 \U0001f620 \U0001f630 \U0001f97a
ASSISTANT: **THINK**[The sentiment of this text is angry]
SYSTEM: OK.
ASSISTANT: **ANSWER**[So the answer is \U0001f620]
SYSTEM: Your answer is correct.
ASSISTANT: **THINK**[I should note the meaning of this emoji for future reference]
SYSTEM: OK.
ASSISTANT: **NOTE**[\U0001f620]: \U0001f620 indicates a sentiment of anger
SYSTEM: case 2.
SYSTEM: When I saw the list of books to read for an acccouting course, I thought "" Oh God how will I ever manaage it! \U0001f616 \U0001f631 \U0001f625 \U0001f92c
ASSISTANT: **THINK**[The sentiment of this text is sadness]
SYSTEM: OK.
ASSISTANT: **ANSWER**[So the answer is \U0001f631]
SYSTEM: Your answer is \U0001f631, however, the correct answer is \U0001f625.
ASSISTANT: **THINK**[I should note the meaning of this emoji for future reference]
SYSTEM: OK.
ASSISTANT: **NOTE**[\U0001f625]: \U0001f625 indicates a sentiment of sadness
SYSTEM: case 3.
SYSTEM: When I dropped a bottle in the state liquor store and everyboby started laughing. \U0001f97a \U0001f922 \U0001f623 \U0001f630
ASSISTANT: The sentiment of this text is shame, so the answer is \U0001f97a
ASSISTANT: **THINK**[The sentiment of this text is shame]
SYSTEM: OK.
ASSISTANT: **ANSWER**[So the answer is \U0001f630]
SYSTEM: Your answer is \U0001f630, however, the correct answer is \U0001f97a.
ASSISTANT: **THINK**[I should note the meaning of this emoji for future reference]
SYSTEM: OK.
ASSISTANT: **NOTE**[\U0001f97a]: \U0001f97a indicates a sentiment of shame | SYSTEM: You are a smart assistant and You need to complete the specified task by talking to the SYSTEM. The dialog between you and SYSTEM will be shown in the form "ASSISTANT:" and "SYSTEM:".

SYSTEM: Your task is to select the emoji that best matches the text.

SYSTEM: Here are several examples.
SYSTEM: case 1.
SYSTEM: When someone stole my bike. \U0001f616 \U0001f620 \U0001f630 \U0001f97a
SYSTEM: Here are several possible relevant experience:
SYSTEM: \U0001f616 indicates a sentiment of guilt
SYSTEM: \U0001f620 indicates a sentiment of anger
SYSTEM: \U0001f630 indicates a sentiment of fear
SYSTEM: \U0001f97a indicates a sentiment of shame
ASSISTANT: **THINK**[The sentiment of this text is angry. Based on these experiences, the answer is \U0001f620]
SYSTEM: OK.
ASSISTANT: **ANSWER**[So the answer is \U0001f620]
SYSTEM: case 2.
SYSTEM: When I saw the list of books to read for an acccouting course, I thought "" Oh God how will I ever manaage it! \U0001f616 \U0001f631 \U0001f625 \U0001f92c
SYSTEM: Here are several possible relevant experience:
SYSTEM: \U0001f616 indicates a sentiment of guilt
SYSTEM: \U0001f631 indicates a sentiment of fear
SYSTEM: \U0001f625 indicates a sentiment of sadness
SYSTEM: \U0001f92c indicates a sentiment of anger
ASSISTANT: **THINK**[The sentiment of this text is sadness. Based on these experiences, the answer is \U0001f625]
SYSTEM: OK.
ASSISTANT: **ANSWER**[So the answer is \U0001f625]
SYSTEM: case 3.
SYSTEM: When I dropped a bottle in the state liquor store and everyboby started laughing. \U0001f97a \U0001f922 \U0001f623 \U0001f630
SYSTEM: Here are several possible relevant experience:
SYSTEM: \U0001f97a indicates a sentiment of shame
SYSTEM: \U0001f922 indicates a sentiment of disgust
SYSTEM: \U0001f623 indicates a sentiment of guilt
SYSTEM: \U0001f630 indicates a sentiment of fear
ASSISTANT: **THINK**[The sentiment of this text is shame. Based on these experiences, the answer is \U0001f97a]
SYSTEM: OK.
ASSISTANT: **ANSWER**[So the answer is \U0001f97a] |
| **Interactive Trajectories** | SYSTEM: Now here is the question:
SYSTEM: Before an accident, my aunt offered me the front seat of the car, next to the driver. I declined, she took that seat and was killed there when the car crashed. \U0001f616 \U0001f621 \U0001f631 \U0001f625
ASSISTANT: **THINK**[The sentiment of this text is regret]
SYSTEM: OK.
ASSISTANT: **ANSWER**[So the answer is \U0001f625]
SYSTEM: Your answer is \U0001f625, however, the correct answer is \U0001f616.
ASSISTANT: **THINK**[I should note the meaning of this emoji for future reference]
SYSTEM: OK.
ASSISTANT: **NOTE**[\U0001f616]: \U0001f616 indicates a sentiment of regret
SYSTEM: OK.
ASSISTANT: END | SYSTEM: Now here is the question:
SYSTEM: When my father and I worked on the house just after he had recovered from illness, I was rather lazy. This meant that he had to work hard and was not very well later. \U0001f633 \U0001f616 \U0001f92e \U0001f62d
SYSTEM: Here are several possible relevant experience:
SYSTEM: \U0001f633 indicates a sentiment of embarrassment
SYSTEM: \U0001f92e indicates a sentiment of disgust
SYSTEM: \U0001f616 indicates a sentiment of regret
SYSTEM: \U0001f62d indicates a sentiment of disappointment
ASSISTANT: **THINK**[The sentiment of this text is regret. Based on these experiences, the answer is \U0001f616]
SYSTEM: OK.
ASSISTANT: **ANSWER**[So the answer is \U0001f616]
SYSTEM: Your answer is correct. |

Figure 7: The example trajectories of ExpNote on the EMOJI Dataset. The left part is a training case while the right part is a corresponding testing case using that training experience. The blue parts are the ExpNote demonstrations (prompts $P_{train}/P_{test}$). The yellow parts are the interactive trajectories between LLM and ExpNote. The sentence highlighted in green is the learned experience.

## Example Trajectory - Training and Testing Cases on the LETS Dataset

| Training Stage | Testing Stage |
|---|---|

**ExpNote Demonstrations**

*Training Stage:*

SYSTEM: You are a smart assistant and You need to complete the specified task by talking to the SYSTEM. The dialog between you and SYSTEM will be shown in the form "ASSISTANT:" and "SYSTEM:".

SYSTEM: Your task is to splice the specified letters together.

SYSTEM: Here are several examples.
SYSTEM: case 1.
SYSTEM: Splice the 6th letter of "turnip", the 2rd letter of "tablet", and the 4th letter of "coin" together.
ASSISTANT: **THINK**[The 6th letter of "turnip" is "p", the 2rd letter of "tablet" is "a", and the 4th letter of "coin" is "n".]
SYSTEM: OK.
ASSISTANT: **ANSWER**[So the spliced letters would be "pan".]
SYSTEM: Your answer is correct.
ASSISTANT: **THINK**[I should note the spelling of these words for future reference]
SYSTEM: OK.
ASSISTANT: **NOTE**[turnip]: "turnip" is composed of "t, u, r, n, i, p". So for the word "turnip", the 1st letter is "t", 2rd letter is "u", 3rd letter is "r", 4th letter is "n", 5th letter is "i", and the 6th letter is "p".
SYSTEM: OK.
ASSISTANT: **NOTE**[tablet]: "tablet" is composed of "t, a, b, l, e, t". So for the word "tablet", the 1st letter is "t", 2rd letter is "a", 3rd letter is "b", 4th letter is "l", 5th letter is "e", and the 6th letter is "t".
SYSTEM: OK.
ASSISTANT: **NOTE**[coin]: "coin" is composed of "c, o, i, n". So for the word "coin", the 1st letter is "c", 2rd letter is "o", 3rd letter is "i", and the 4th letter is "n".
SYSTEM: OK.
SYSTEM: case 2.
SYSTEM: Splice the 4th letter of "manual", the 3th letter of "represent", and the 4th letter of "official" together.
ASSISTANT: **THINK**[The 4th letter of "manual" is "u", the 3th letter of "represent" is "r", and the 4th letter of "official" is "i".]
SYSTEM: OK.
ASSISTANT: **ANSWER**[So the spliced letters would be "uri".]
SYSTEM: Your answer is "uri", however, the correct answer is "upi".
ASSISTANT: **THINK**[I should note the spelling of these words for future reference]
SYSTEM: OK.
ASSISTANT: **NOTE**[manual]: "manual" is composed of "m, a, n, u, a, l". So for the word "manual", the 1st letter is "m", 2rd letter is "a", 3rd letter is "n", 4th letter is "u", 5th letter is "a", and the 6th letter is "l".
SYSTEM: OK.
ASSISTANT: **NOTE**[represent]: "represent" is composed of "r, e, p, r, e, s, e, n, t". So for the word "represent", the 1st letter is "r", 2rd letter is "e", 3rd letter is "p", 4th letter is "r", 5th letter is "e", 6th letter is "s", 7th letter is "e", 8th letter is "n", and the 9th letter is "t".]
SYSTEM: OK.
ASSISTANT: **NOTE**[official]: "official" is composed of "o, f, f, i, c, i, a, l". So for the word "official", the 1st letter is "o", 2rd letter is "f", 3rd letter is "f", 4th letter is "i", 5th letter is "c", 6th letter is "i", 7th letter is "a", and the 8th letter is "l".]
SYSTEM: OK.

*Testing Stage:*

SYSTEM: You are a smart assistant and You need to complete the specified task by talking to the SYSTEM. The dialog between you and SYSTEM will be shown in the form "ASSISTANT:" and "SYSTEM:".

SYSTEM: Your task is to splice the specified letters together.

SYSTEM: Here are several examples.
SYSTEM: case 1.
SYSTEM: Splice the 4th letter of "manual", the 3th letter of "represent", and the 4th letter of "official" together.
SYSTEM: No relevant experience.
ASSISTANT: **THINK**[Seems there is no relevant experience about this question, I need to rely on myself]
SYSTEM: OK.
ASSISTANT: **THINK**[The 4th letter of "manual" is "u", the 3th letter of "represent" is "p", and the 4th letter of "official" is "i".]
SYSTEM: OK.
ASSISTANT: **ANSWER**[So the spliced letters would be "upi".]
SYSTEM: Your answer is correct.
SYSTEM: case 2.
SYSTEM: Splice the 6th letter of "turnip", the 3rd letter of "tablet", and the 4th letter of "coin" together.
SYSTEM: Here are several relevant experience:
SYSTEM: "turnip" is composed of "t, u, r, n, i, p". So for the word "turnip", the 1st letter is "t", 2rd letter is "u", 3rd letter is "r", 4th letter is "n", 5th letter is "i", and the 6th letter is "p".
SYSTEM: "coin" is composed of "c, o, i, n". So for the word "coin", the 1st letter is "c", 2rd letter is "o", 3rd letter is "i", and the 4th letter is "n".
ASSISTANT: **THINK**[Based on these experiences, the first letter is "p", the third letter is "n"]
SYSTEM: OK.
ASSISTANT: **THINK**[And the the 3rd letter of "tablet" is "b", so the second letter is "b"]
SYSTEM: OK.
ASSISTANT: **ANSWER**[So the spliced letters would be "pbn".]
SYSTEM: Your answer is correct.

**Interactive Trajectories**

*Training Stage:*

SYSTEM: Now here is the question:
SYSTEM: Splice the 7th letter of "greeting", the 5th letter of "matter", and the 4th letter of "reed" together.
ASSISTANT: **THINK**[The 7th letter of "greeting" is "n", the 5th letter of "matter" is "e", and the 4th letter of "reed" is "d".]
SYSTEM: OK.
ASSISTANT: **ANSWER**[So the spliced letters would be "ned".]
SYSTEM: Your answer is correct.
ASSISTANT: **THINK**[I should note the spelling of these words for future reference]
SYSTEM: OK.
ASSISTANT: **NOTE**[greeting]: "greeting" is composed of "g, r, e, e, t, i, n, g". So for the word "greeting", the 1st letter is "g", 2rd letter is "r", 3rd letter is "e", 4th letter is "e", 5th letter is "t", 6th letter is "i", 7th letter is "n", and the 8th letter is
SYSTEM: OK.
ASSISTANT: **NOTE**[matter]: "matter" is composed of "m, a, t, t, e, r". So for the word "matter", the 1st letter is "m", 2rd letter is "a", 3rd letter is "t", 4th letter is "t", 5th letter is "e", and the 6th letter is "r".
SYSTEM: OK.
ASSISTANT: **NOTE**[reed]: "reed" is composed of "r, e, e, d". So for the word "reed", the 1st letter is "r", 2rd letter is "e", 3rd letter is "e", and the 4th letter is "d".
SYSTEM: OK.

*Testing Stage:*

SYSTEM: Now here is the question:
SYSTEM: Splice the 3th letter of "story", the 6th letter of "possess", and the 1st letter of "reed" together.
SYSTEM: Here are several possible relevant experience:
SYSTEM: "possess" is composed of "p, o, s, s, e, s, s". So for the word "possess", the 1st letter is "p", 2rd letter is "o", 3rd letter is "s", 4th letter is "s", 5th letter is "e", 6th letter is "s", and the 7th letter is "s".
SYSTEM: "reed" is composed of "r, e, e, d". So for the word "reed", the 1st letter is "r", 2rd letter is "e", 3rd letter is "e", and the 4th letter is "d".
SYSTEM: "story" is composed of "s, t, o, r, y". So for the word "story", the 1st letter is "s", 2rd letter is "t", 3rd letter is "o", 4th letter is "r", and the 5th letter is "y".
ASSISTANT: **THINK**[Based on these experiences, the 3rd letter of "story" is "o", the 6th letter of "possess" is "s", and the 1st letter of "reed" is "r".]
SYSTEM: OK.
ASSISTANT: **ANSWER**[So the spliced letters would be "osr".]
SYSTEM: Your answer is correct.

Figure 8: The example trajectories of ExpNote on the LETS Dataset. The left part is a training case while the right part is a corresponding testing case using that training experience. The blue parts are the ExpNote demonstrations (prompts $P_{train}/P_{test}$). The yellow parts are the interactive trajectories between LLM and ExpNote. The sentence highlighted in green is the learned experience.