# OpenReview forum: "ExpNote: Black-box Large Language Models are better Task Solvers with Experience Notebook"
_EMNLP/2023/Conference — EMNLP 2023 Findings_

### Official Review · Reviewer_ZAJN · 2023-08-05

**Soundness:** 2

**Excitement:**

2: Mediocre: This paper makes marginal contributions (vs non-contemporaneous work), so I would rather not see it in the conference.

**Missing References:**

[1] Making LLMs Better Reasoners with Step-Aware Verifier by Li et al. 2022 presents a technique for finding more reliable CoT reasoning paths. It has strong results on CLUTTR.

[2] Compositional Semantic Parsing with LLMs by Drozdov et al. 2022 uses demonstrations that include their pre-computed CoT. Also, the demonstration retrieval involves template-like steps that have some similarity with experiences used in ExpNote.

[3] Faithful CoT (Lyu et al) shows that adding explanations to CoT is beneficial.

**Paper Topic And Main Contributions:**

The authors present a prompting that utilizes demonstration-like retrieval. I call it "demonstration-like" because what is included in the prompt is derived from a demonstration (called relevant experience). Although a potentially novel technique, I found the method is talked about quite abstractly and the paper is missing details that would be necessary for replication. In addition, although the paper includes multiple ablations there are not sufficient baselines to know if this is an overall promising technique. As an example, at least one prompting baseline is missing for CLUTTR which greatly outperforms ExpNote.

**Questions For The Authors:**

Did you consider retrieving relevant demonstrations rather than only using a fixed set of demonstrations? In other words, is it clear that using demonstration as "experiences" is more important than simply using the raw demonstrations?

**Reasons To Accept:**

Strength 1. It is interesting to include "experiences" which are derived from demonstrations rather than demonstrations themselves. That being said, this has been done before with chain-of-thought.

Strength 2. The paper surprisingly shows that using experiences where the model failed (called negatives) is very helpful. Although, on closer inspection, this may be because the failure rate is high and that too much data is excluded when excluding failures.

Strength 3. ExpNote substantially outperforms ablations and two baselines (TeachMe and Reflexion). Although some important prompting baselines may be missing.

**Reasons To Reject:**

Reject 1. There are probably not enough details to reproduce the work without guessing at multiple steps. For example, it is not clear what is a "word-based retriever".

Reject 2. ExpNote probably needs negatives for good performance because so many of the initial predictions are negatives. The positives are only 70% in the best case (EMOJI) and can be less than 50% on some data. Plus the data is somewhat small to begin with, making the inclusion of as much of the train data as possible important.

Reject 3. It seems the baselines are not particularly strong. For example, DIVERSE can achieve > 90% accuracy on CLUTTR [1].

**Reproducibility:**

3: Could reproduce the results with some difficulty. The settings of parameters are underspecified or subjectively determined; the training/evaluation data are not widely available.

**Reviewer Confidence:**

5: Positive that my evaluation is correct. I read the paper very carefully and I am very familiar with related work.

**Typos Grammar Style And Presentation Improvements:**

Why are no decimals shown in Table 1/2? Is the full data being used or is this a sample of 100 test cases?

---

> ### Author Rebuttal · Authors · 2023-08-29
>
> ### Thank you very much for your review. Here are our responses to your questions.
> #### Paper Topic And Main Contributions:
> - Q: The authors present a prompting that utilizes demonstration-like retrieval. I call it "demonstration-like" because what is included in the prompt is derived from a demonstration (called relevant experience). Although a potentially novel technique, I found the method is talked about quite abstractly and the paper is missing details that would be necessary for replication. In addition, although the paper includes multiple ablations there are not sufficient baselines to know if this is an overall promising technique. As an example, at least one prompting baseline is missing for CLUTTR which greatly outperforms ExpNote.
> - A:
>     - Our proposed ExpNote framework and methods such as demonstration have similar functionality in prompting LLM to generalize from known information to new and unseen instances. However, our method has unique features in extracting, storing, recalling, and utilizing these aspects. We have designed a set of action prompts that interact with external experience bases, combining the advantages of reflection-based and memory-based approaches from a new perspective.
>     - ExpNote fully automates the learning and induction of experiences from the training set, providing prompts with a fixed number of experiences, saving on the number of samples and storage space required for demonstration using input-output pairs, and improving generalization efficiency.
>     - We have provided clear code to support the reproducibility of our work. We have applied the framework on datasets in different fields and tasks, demonstrating its universality and robustness.
>
> #### Reasons To Accept:
> - Q1: It is interesting to include "experiences" which are derived from demonstrations rather than demonstrations themselves. That being said, this has been done before with chain-of-thought.
> - A1: We have constructed a new approach to guide LLM in inducing experience from task training sets and enabling recall of these experiences during testing. Unlike COT, our approach focuses more on extracting and storing experiential information from task-specific perspectives for better application in subsequent inference processes.
> - Q2: The paper surprisingly shows that using experiences where the model failed (called negatives) is very helpful. Although, on closer inspection, this may be because the failure rate is high and that too much data is excluded when excluding failures.
> - A2: The experiences from negative examples are indeed important for our framework. But this is not because the failure rate is high and that too much data is excluded when excluding failures. In fact, as shown in Table 2, in 3 out of 4 datasets, the positive experiences are more than the negative experiences (The numbers in the small bracket are numbers of experiences stored in the memory). **We think the true reason is that the data from model failures contains valuable information for specific tasks, which may include errors in the model's reasoning and judgment in different situations. ExpNote can help the model better understand and adapt to new situations.**
> - Q3: ExpNote substantially outperforms ablations and two baselines (TeachMe and Reflexion). Although some important prompting baselines may be missing.
> - A3: Compared to methods such as CoT, reflection-based, and memory-based methods, ExpNote's performance is outstanding, and more importantly, the framework is fully automated without the need for human intervention. Regarding the issue about other baselines, we will explain it in the following reply.
> #### Reasons To Reject:
> - Q1: There are probably not enough details to reproduce the work without guessing at multiple steps. For example, it is not clear what is a "word-based retriever".
> - A1: We highlighted several main contributions of the ExpNote framework in the paper, including the design of a fully automated training paradigm that integrates the advantages of reflection and memory, and the design of multiple action instructions that interact with external experience bases. Due to the limitations of the paper, we did briefly describe some details, but through our publicly available code implementation, Readers can have a deeper understanding of each step. For example, for a word-based retriever, in the 3.2 Interaction Commands section of the paper, we explicitly mentioned "using the question arg as the search query". Specifically, it retrieves experiences by matching words in the query and the key of experience and then selecting the k most relevant experiences.
> - Q2: ExpNote probably needs negatives for good performance because so many of the initial predictions are negatives. The positives are only 70% in the best case (EMOJI) and can be less than 50% on some data. Plus the data is somewhat small to begin with, making the inclusion of as much of the train data as possible important.
> - A2:
>     - Our work found that negative data may contain useful information for specific tasks, and ExpNote is able to fully utilize these samples in both training and testing processes. By storing experiences in external memory, the model can retrieve and utilize these experiences in different tasks and situations, thereby enhancing its generalization ability. **We think leveraging the negative cases during training to generate experiences for testing is a novel idea in our paper instead of a weakness.**
>     - In order to analyze the impact of experience types on performance, we have conducted relevant experiments and trained our method on only correctly answered sample datasets, which can still significantly outperform the baseline, as shown in Table 2.
>     - Regarding the issue of data size, our experimental results indicate that even with relatively small initial data sizes, ExpNote is still able to continuously learn experience during the training process as the sample size increases (Figure 3), and has achieved significant performance improvements on multiple tasks.
> - Q3: It seems the baselines are not particularly strong. For example, DIVERSE can achieve > 90% accuracy on CLUTTR [1].
> - A3: Thank you for pointing out. While DIVERSE performs well on CLUTTR, we believe it is not suitable for comparison with our work.
>     - There are several reasons for this:
>         1. DIVERSE has made multiple modifications to the CLUTTR benchmark, including appending candidate answers and constructions after the question. Their experiments show that without these candidate answer sequences, the accuracy of LLM does not exceed 50%, while we achieve an accuracy of 61% without additional prompts.
>         2. DIVERSE incorporates manually constructed reasoning paths into every question-answer pair in the training set, requiring a significant amount of manual annotation. In contrast, ExpNote only requires a small number of prompts to automatically induce and generalize from the training set.
>         3. DIVERSE uses self-consistency decoding, generating different reasoning paths and conducting majority voting, while ExpNote uses greedy decoding, where the large model samples the most likely word from the vocabulary. In Faithful CoT (Lyu et al.), these two types of methods are compared separately.
>     - Additionally, in comparison to the methodology of the comparative approach, ExpNote's advantage lies in its simple experience retrieval during the testing process and the complete automation of the entire training process without human intervention.
>         1. DIVERSE attempts to generate diverse prompts, exploring different reasoning paths for the same question, and then filters out incorrect answers based on a weighted voting scheme, verifying each reasoning step individually.
>         2. This method has high complexity and requires human involvement in various stages. For example, it involves the need for more annotated data with well-annotated reasoning paths to construct diverse prompts. It also requires a training dataset for supervised validators and relies on human evaluation to measure the quality of intermediate steps in the reasoning paths.
> #### Questions For The Authors:
> - Q: Did you consider retrieving relevant demonstrations rather than only using a fixed set of demonstrations? In other words, is it clear that using demonstration as "experiences" is more important than simply using the raw demonstrations?
> - A: Thank you for your question. We have already taken this into consideration. In line 229, we introduced a variant case of the ExpNote framework, which recalls the original relevant Q&A pairs instead of learning experiences. The experiment shows that ExpNote's use of reflective ability to summarize experiences makes LLM perform better in specific downstream tasks than simply using case or demonstration, and the effect is significant.
> #### Missing References:
> - Q1: [1] Making LLMs Better Reasoners with Step-Aware Verifier by Li et al. 2022 presents a technique for finding more reliable CoT reasoning paths. It has strong results on CLUTTR.
> - A1: DIVERSE is not suitable for comparison with our method, as detailed in Q3's answer.
> - Q2: [2] Compositional Semantic Parsing with LLMs by Drozdov et al. 2022 uses demonstrations that include their pre-computed CoT. Also, the demonstration retrieval involves template-like steps that have some similarity with experiences used in ExpNote.
> - A2: The experience in ExpNote is an abstract knowledge or rule related to a specific task. This literature proposes the least to most method to perform multi-step semantic analysis of the problem, decompose the sub-questions, and output the answers to the sub-questions to prompt the final answer. The information obtained through retrieval decomposition is a 'sub problem', which is significantly different from the experience in ExpNote.
> - Q3: [3] Faithful CoT (Lyu et al) shows that adding explanations to CoT is beneficial.
> - A3: ExpNote mainly compares reflection-based and memory-based methods, and may have missed out on Faithful CoT. The main difference between the two is that Faithful CoT manually sets fixed enhanced inference prompts for each task, while ExpNote automatically induces abstract knowledge or rules to obtain more types of experience for use, and can be transferred to tasks with similar domain knowledge.
> Thank you very much for your suggestion. We will consider adding it in the revised version.
> #### Typos Grammar Style And Presentation Improvements:
> - Q: Why are no decimals shown in Table 1/2? Is the full data being used or is this a sample of 100 test cases?
> - A: The reason for the lack of decimal points is indeed because various methods were tested on 100 test samples. Due to the size limitations of the datasets themselves, we used samples of the order of 10^2 for training and testing. In fact, a large amount of related work is also tested using samples of similar magnitude, such as TeachMe (OBQA, 500), ReAct (ALFWorld, 134; WebShop, 500), Reflexion (consistent with ReAct). We believe that the ExpNote method emphasizes the integration of experience and external memory, and performance improvement in different fields and tasks can reflect this.

---

### Official Review · Reviewer_RD22 · 2023-08-06

**Soundness:** 3

**Excitement:**

4: Strong: This paper deepens the understanding of some phenomenon or lowers the barriers to an existing research direction.

**Paper Topic And Main Contributions:**

This paper presents a framework called Experience Notebook or ExpNote for providing external and dynamic memories for large language models (LLM).  The authors design a set of action prompts to enable the reading and writing with the external memory. During training,  ExpNote will automatically generalize task-specific experiences to external memory using reflection thinking. During testing, ExpNote retrieves related memory for supporting the LLM’s decisions. Experiments on four tasks show that the proposed methods are much better than the baseline models including Chain-of-Thought,  Reflexion and TeachMe.

**Questions For The Authors:**

Please see point 1 in Weaknesses.

**Reasons To Accept:**

1. The research topic is quite interesting and would draw a lot of attention from the community.
2. The proposed method is reasonable and effective on multiple benchmarks.
3. The authors design sufficient ablation studies to show the effectiveness of the proposed method.
4. The code is released.


**Reasons To Reject:**

1. The main problem of this paper is the unclear writing.  It is not easy to follow the whole process without knowing the clear details.  For example,  to store the task-specific experience,  key-value pairs are necessary. How to generate such key and value pairs are not fully described.  For another example, how to generalize a training instance to an abstract experience?  Is this done by heuristic rules for specific format data?  When there are multiple experiences recalled during a testing phrase, how to decide which one to be used?   After reading the codes, these questions become clearer for me.  However, it should be clearly described in the original texts. I suggest that the example prompts in the dataloader should be appended in the appendix.

2. Why do you choose these four datasets?  The previous three datasets are not well-known for LLM evaluations. And the last one is a synthetic dataset.  Are there any specific reasons?


**Reproducibility:**

4: Could mostly reproduce the results, but there may be some variation because of sample variance or minor variations in their interpretation of the protocol or method.

**Reviewer Confidence:**

4: Quite sure. I tried to check the important points carefully. It's unlikely, though conceivable, that I missed something that should affect my ratings.

---

> ### Author Rebuttal · Authors · 2023-08-29
>
> ### Thank you very much for your review. Here are our responses to your questions.
> - Q1: The main problem of this paper is the unclear writing. It is not easy to follow the whole process without knowing the clear details.
>   - For example, to store the task-specific experience, key-value pairs are necessary. How to generate such key and value pairs are not fully described.
>   - For another example, how to generalize a training instance to an abstract experience? Is this done by heuristic rules for specific format data? When there are multiple experiences recalled during a testing phrase, how to decide which one to be used?
>   - After reading the codes, these questions become clearer for me.
>   - However, it should be clearly described in the original texts. I suggest that the example prompts in the dataloader should be appended in the appendix.
> - A1: Thank you very much for your advice. We have recognized these points. As we prefer to present the ideas and experimental methods of the ExpNote framework, due to space limitations, we have appropriately simplified the prompts and operating procedures related to data loading. Our code clearly presents how ExpNote generates key values to store task-specific experiences, how to generalize training examples into abstract experiences, and how to recall and use experiences. We will add relevant implementation details to the appendix in the revised version.
> - Q2: Why do you choose these four datasets? The previous three datasets are not well-known for LLM evaluations. And the last one is a synthetic dataset. Are there any specific reasons?
> - A2: Thank you very much for your interest in dataset selection. When selecting these four datasets, we considered some specific factors that cover tasks in different fields. CLUTRR is a classic dataset for inductive reasoning and is often used for comparison in related COT research. METS-CoV and EMOJI are relatively difficult downstream task datasets that often require fine-tuning to achieve better performance. In addition, currently, the symbol inference dataset is mainly used to test the zero-shot ability of LLM, and no training set has been designed. Therefore, we constructed LETS to evaluate ExpNote's ability to enhance LLM's learning and generalization in symbol inference tasks. We verify our proposed framework on these datasets for different tasks to ensure that the applicability and robustness of the method are comprehensively evaluated.

---

### Official Review · Reviewer_CVaE · 2023-08-11

**Soundness:** 2

**Excitement:**

3: Ambivalent: It has merits (e.g., it reports state-of-the-art results, the idea is nice), but there are key weaknesses (e.g., it describes incremental work), and it can significantly benefit from another round of revision. However, I won't object to accepting it if my co-reviewers champion it.

**Paper Topic And Main Contributions:**

This paper proposes the ExpNote (Experience Notebook) framework, a novel LLM prompting method. ExpNote reflects experiences and store them as notes in an external memory in the training, and retrieves the experiences in the inference to help the LLM better solve unfamiliar tasks. The authors compared their proposed method with three strong baselines involving Chain-of-Thought (CoT), memory-based and reflection-based prompting methods. ExpNote outperforms these baselines on four different types of tasks in inductive reasoning, medical entity recognition, text-to-emoji prediction and letter splicing. They also performed an ablation study with several variants of ExpNote. The main contribution of this short paper is to propose a novel and effective framework ExpNote for LLM in-context learning.

**Reasons To Accept:**

* This paper proposes a novel and effective framework for LLM in-context learning and achieves better performance against strong baselines.
* The paper was written well and fluently, easy to follow. The framework figures are illusitrative and help the readers know the main idea easily.

**Reasons To Reject:**

* All experiments were performed on a very limited of test sets (100 cases) and train sets (minimal 100 or 200 cases according to Appendix B), potentially causing the risk of randomness in terms of the results.
* The ExpNote framework was only tested with one LLM, i.e., ChatGPT (which was even not mentioned in the main part of the paper, just occuring in the Appendix), which is not solid and persuasive enough.
* The main contribution of this paper is a little incremental to previous work like memory-based and reflection-based prompting methods.

**Reproducibility:**

5: Could easily reproduce the results.

**Reviewer Confidence:**

3: Pretty sure, but there's a chance I missed something. Although I have a good feel for this area in general, I did not carefully check the paper's details, e.g., the math, experimental design, or novelty.

---

> ### Author Rebuttal · Authors · 2023-08-29
>
> ### Thank you very much for your review. Here are our responses to your questions.
> - Q1: All experiments were performed on a very limited of test sets (100 cases) and train sets (minimal 100 or 200 cases according to Appendix B), potentially causing the risk of randomness in terms of the results.
> - A1: Due to the size limitations of the datasets themselves, we used samples of the order of 10^2 for training and testing. **In fact, a large amount of related work is also tested using samples of similar magnitude, such as TeachMe (OBQA, 500), ReAct (ALFWorld, 134; WebShop, 500), Reflexion (consistent with ReAct).** Our proposed LLM in-context learning framework can induce experience from the training set and help generalize it to the test set. It can also demonstrate sustained learning ability and good performance in a small number of samples. The dataset contains both balanced positive and negative examples, fully reflecting the possible distribution of samples in the task dataset. The superior performance of the experiment on multiple different tasks and datasets can demonstrate the universality of the method.
> - Q2: The ExpNote framework was only tested with one LLM, i.e., ChatGPT (which was even not mentioned in the main part of the paper, just occurring in the Appendix), which is not solid and persuasive enough.
> - A2: Considering the good performance of ChatGPT (gpt-3.5 turbo) on various NLP tasks, in order to highlight the universality of the ExpNote framework, we are currently verifying the effectiveness of the method on this LLM. The current experimental results can provide inspiration for other similar models. Thank you for your feedback. We will explore applying other LLMs to the framework of this article in the future.
> - Q3: The main contribution of this paper is a little incremental to previous work like memory-based and reflection-based prompting methods.
> - A3: Exploring the boundaries of LLM capabilities is a novel frontier direction. The ExpNote framework proposed in this paper brings a new perspective to the research direction of the entire LLM prompting method, creatively integrating the advantages of memory-based and reflection-based prompting methods, enabling ExpNote to summarize abstract knowledge and rules from correct or incorrect responses, enabling the model to better handle unfamiliar tasks and unseen samples. By conducting experiments on multiple tasks, it was demonstrated that ExpNote performs significantly better than a single method in different scenarios. The experimental results on tasks such as inductive reasoning, entity recognition, and symbolic reasoning also indicate that our method can significantly improve the performance of large models by 20.5% compared to CoT; Compared to other methods based solely on memory and reflection, there has also been a significant improvement.

---

### Official Review · Reviewer_r6eq · 2023-08-11

**Soundness:** 3

**Excitement:**

3: Ambivalent: It has merits (e.g., it reports state-of-the-art results, the idea is nice), but there are key weaknesses (e.g., it describes incremental work), and it can significantly benefit from another round of revision. However, I won't object to accepting it if my co-reviewers champion it.

**Paper Topic And Main Contributions:**

The authors propose a compelling framework to incorporate LLM experiences from training data to improve LLM performance on test instances. Experiments are conducted on 4 different datasets on ChatGPT (gpt-3.5-turbo). The authors also propose a new dataset LETS to evaluate the symbolic reasoning capabilities of LLM.

**Questions For The Authors:**

Line 35 is not clear.

**Reasons To Accept:**

- Empirically, it is evident that the proposed approach is better than existing reflection/memory-based methods.
- Table 2 clearly shows that the LLM equally benefits from both positive and negative experiences.

**Reasons To Reject:**

- How the task-specific experiences are learned for example line 147: "if A has a daughter B, and A is married to C, then B is also the daughter of C". Are these experiences part of the data? If not, the poor quality of these experiences can negatively impact the testing.
- What are in-context learning examples during testing? Are these examples with the experience recall or just input-output pairs?
- What retriever is used? How the relevant experiences are selected? I do not find any information in the paper.
- Why the author thinks recent reflection-based methods (Shinn et al., 2023; Madaan et al., 2023; Akyürek et al., 2023) do not generalize to other instances or tasks?

**Reproducibility:**

4: Could mostly reproduce the results, but there may be some variation because of sample variance or minor variations in their interpretation of the protocol or method.

**Reviewer Confidence:**

4: Quite sure. I tried to check the important points carefully. It's unlikely, though conceivable, that I missed something that should affect my ratings.

---

> ### Author Rebuttal · Authors · 2023-08-29
>
> ### Thank you very much for your review. Here are our responses to your questions.
> - Q1: How the task-specific experiences are learned for example line 147: "if A has a daughter B, and A is married to C, then B is also the daughter of C". Are these experiences part of the data? If not, the poor quality of these experiences can negatively impact testing.
> - A1: These experiences are not part of the original data, but are automatically generated by the LLMs through few-shot prompting. By constructing example prompts, LLM learns task-specific experiences that can be generalized for other testing cases (specific prompts can be found at expnote_train in ./dataloader/<task>.py). High-quality experience is indeed important. For specific tasks, heuristic rules and examples are used to guide the ExpNote framework in automatically summarizing experience. By extracting experience from positive/negative samples in the dataset and generalizing it to the test set, relevant experiments have shown that the quality of experience obtained by this method is better than simply using examples for demonstration, and it is also more robust than reflection-based methods.
> - Q2: What are in-context learning examples during testing? Are these examples with the experience recall or just input-output pairs?
> - A2: In-context learning examples during testing include experience recall. The LLMs learn to reason based on recall experience during testing through the ability of in-context learning (specific prompts can be found at expnote_test in ./dataloader/<task>.py)
> - Q3: What retriever is used? How the relevant experiences are selected? I do not find any information in the paper.
> - A3: As stated in Appendix B, we used a keyword-based retriever to retrieve experience by matching words in the query and the key of experience, and then selecting the k most relevant experiences. In this paper, we mainly emphasize the overall idea and process of the ExpNote framework, without overly considering the implementation methods of different retrievers. In the future, we will compare the impact of different retrievers on the model's performance.
> - Q4: Why the author thinks recent reflection-based methods (Shinn et al., 2023; Madaan et al., 2023; Akyürek et al., 2023) do not generalize to other instances or tasks?
> - A4: Methods that are solely based on reflection only generate case-level specific experience. For example, for a testing case in ALFWorld (a dataset used in Reflexion), the reflection is ' In the next trial, I will go to desk 1, find the lamp, then look for the mug and examine it with the desklamp '. Therefore, these reflections can only be used to prompt the next trial of the same case, and can not be generalized to other test cases. In contrast, by storing and then retrieving, ExpNote can help LLM further think and more comprehensively utilize the information in the training set that is helpful for specific tasks.
> - Q: Line 35 is not clear.
> - A: Regarding the black box model of ChatGPT, we used OpenAI's gpt-3.5 turbo interface, as mentioned in Appendix B.

---

### Meta-Review · Area_Chair_WS2E · 2023-09-17

**Recommendation:** 3

**Metareview:**

Reviewers agree this work offers an interesting and topical method for leveraging training data "notes"/experience at inference-time. Though some are concerned its contributions may be incremental, it does introduce nice augmentations on existing work for a short paper. Its results indicate strong potential, however, there are also universal concerns with understanding crucial details of the method and experimental setup, as well as missing related work. After significant discussion, I hope the authors will heed the advice of the reviewers and agree to (a) fill in missing details from their methodology, (b) add the requested modifications to the related work, and (c) augment the discussion with their additional experiments and carefully contextualize their quantitative results given the small train/test sizes (which also appears in prior work). With these changes the work will be much improved.

---

### Decision · Program_Chairs · 2023-10-07

**Decision:**

Accept-Findings

**Comment:**

Reviewers agree this work offers an interesting and topical method for leveraging training data "notes"/experience at inference-time. Though some are concerned its contributions may be incremental, it does introduce nice augmentations on existing work for a short paper. Its results indicate strong potential, however, there are also universal concerns with understanding crucial details of the method and experimental setup, as well as missing related work. After significant discussion, I hope the authors will heed the advice of the reviewers and agree to (a) fill in missing details from their methodology, (b) add the requested modifications to the related work, and (c) augment the discussion with their additional experiments and carefully contextualize their quantitative results given the small train/test sizes (which also appears in prior work). With these changes the work will be much improved.